# Current Landscape of Targeted Therapy in Hormone Receptor-Positive and HER2-Negative Breast Cancer

**Samitha Andrahennadi** [1] , **Amer Sami** [1,2] , **Mita Manna** [1,2] , **Mehrnoosh Pauls** [1,2] **and Shahid Ahmed** [1,2,*]

1    College of Medicine, University of Saskatchewan, Saskatoon, SK S7N 5E5, Canada;
     sma936@mail.usask.ca (S.A.); amer.sami@saskcancer.ca (A.S.); mita.manna@saskcancer.ca (M.M.);
     Mehrnoosh.Pauls@saskcancer.ca (M.P.)
2    Saskatoon Cancer Center, Saskatchewan Cancer Agency, University of Saskatchewan, 20 Campus Drive,
     Saskatoon, SK S7N 4H4, Canada
*    Correspondence: shahid.ahmed@saskcancer.ca; Tel.: +1-306-655-2710; Fax: +1-306-655-0633

**Abstract:** *Background*: Hormone receptor-positive and HER2-negative breast cancer (HR + BC) is the most prevalent breast cancer. Endocrine therapy is the mainstay of treatment, however, due to the heterogeneous nature of the disease, resistance to endocrine therapy is not uncommon. Over the past decades, the emergence of novel targeted therapy in combination with endocrine therapy has shown improvement in outcomes of HR + BC. This paper reviews available data of targeted therapy and the results of pivotal clinical trials in the management of HR + BC. *Methods*: A literature search in PubMed and Google Scholar was performed using keywords related to HR + BC and targeted therapy. Major relevant studies that were presented in international cancer research conferences were also included. *Results*: Endocrine therapy with tamoxifen and aromatase inhibitors are backbone treatments for women with early-stage HR + BC leading to a significant reduction in mortality. They can also be used for primary prevention in women with a high risk of breast cancer. Preliminary data has shown the efficacy of adjuvant cyclin-dependent kinase (CDK) 4/6 inhibitor, abemaciclib, in high-risk disease in combination with aromatase inhibitors. For most women with advanced HR + BC, endocrine therapy is the primary treatment. Recent evidence has shown that the use of CKD 4/6 inhibitors, mTOR inhibitors, and PI3K inhibitors in combination with endocrine therapy has been associated with better outcomes and delays initiation of chemotherapy. Several novel agents are under study for HR + BC. *Discussion*: Targeted treatment options for HR + BC have evolved. The future of overcoming resistance to targeted therapy, novel compounds, and predictive markers are key to improving HR + BC outcomes.

**Keywords:** breast cancer; hormone receptor-positive breast cancer; estrogen receptor-positive-breast cancer; systemic therapy; targeted therapy; antiestrogen therapy; selective estrogen receptor modulator; selective estrogen receptor degraders; cyclin-dependent kinases 4 and 6 inhibitors; aromatase inhibitors



## 1. Introduction

Breast cancer (BC) is the leading cause of cancer-related death in women [1,2]. However, over the past three decades, the outcomes of women with BC have improved [3]. Hormone receptor-positive HER2-negative breast cancer (HR + BC) is the most prevalent BC characterized by estrogen or progesterone receptor (ER/PR) positive HER2-negative malignant cells comprised of about 60–70% of all BC [2,4]. Antiestrogen therapy is the first targeted therapy approved in the management of HR + BC [3,5,6]. Subsequently, a better understanding of biology and the underlying mechanism of HR + BC led to the discovery of other novel targeted agents that in combination with antiestrogen therapy have shown promising outcomes in women with HR + BC [6]. This review focuses on the current status of targeted therapy in HR + BC.

## 2. Literature Search

A literature search using PubMed and Google Scholar was performed. The following keywords were used: "breast cancer", or "hormone receptor-positive breast cancer", or "estrogen receptor-positive-breast cancer," and "systemic therapy", "targeted therapy", "antiestrogen therapy", "selective estrogen receptor modulator", "selective estrogen receptor degraders", "cyclin-dependent kinases 4 and 6 inhibitors", and "aromatase inhibitors." The studies that were published in English prior to December 2020 and limited to humans were assessed. In addition, studies that were presented in the major international cancer research conferences were included.

## 3. Antiestrogen (Endocrine) Therapy

Estrogens are steroidal female sex hormones that are produced from androgens by aromatase enzymes [7]. They increase the risk of breast, endometrial, and ovarian cancer [8]. HR + BC cells rely on estrogen for their growth and proliferation [8,9]. Estrogens mediate their effect by binding and activating nuclear receptor ERα and ERβ and thereby promoting relevant gene transcription [7,10]. Antiestrogen treatments in both early and advanced HR + BC have been shown to be associated with improvement in overall survival (OS) [11–15]. Currently, three classes of antiestrogen treatments are available for patients with HR + BC (Table 1): selective estrogen receptor modulators (SERMs), aromatase inhibitors (AIs), and selective estrogen receptor degraders (SERDs) [13,16]. In the following sections, we review the data on the efficacy of antiestrogen therapy in BC.

**Table 1.** Key targeted therapies that are approved in the management of HR + BC.

| Targeted Agent | Mechanism of Action |
| --- | --- |
| Selective estrogen receptor modulators (SERMs): tamoxifen, raloxifene and toremifene | Compete with estrogen to bind to estrogen receptors and based on target tissue act differentially on estrogen receptor as antagonist or partial agonist. |
| Aromatase inhibitors: non-steroidal anastrozole and letrozole, and steroidal exemestane | Inactivate aromatase enzyme that converts androgens to estrogens and thereby suppress plasma estrogen level. |
| Selective estrogen receptor degrader (SERD): fulvestrant | Pure estrogen receptor antagonist, exerting selective ER downregulation, and competitively binding to the ER |
| Cyclin D Kinase 4/6 inhibitors: Palbociclib, ribociclib and abemaciclib | Inhibit CDK 4/6 that are responsible for phosphorylation and inactivation of retinoblastoma protein. |
| PI3K/Akt/mTOR (PAM) pathway inhibitors mTOR inhibitor: everolimus PI3K inhibitor: alpelisib | Inhibits PI3K/Akt/mTOR, which are key mediators of the cell cycle, and are often overactive in breast cancer. |

### 3.1. Selective Estrogen Receptor Modulators (SERMs)

SERMs are a group of drugs that compete with estrogen to bind to the ER and exert either an antiestrogenic effect or an estrogenic effect based on the target tissue [16]. Tamoxifen is the most studied SERM in BC. Raloxifene and toremifene are the other SERMS that have been evaluated in the prevention or treatment of BC [11–13,17]. Tamoxifen inhibits the actions of endogenous estrogen in normal breast and breast cancer cells but acts like estrogen in other tissues, including bone, uterus, liver, and coagulation proteins. The antiestrogen effect of tamoxifen results in reduction in the risk of invasive and in situ breast cancer. It has shown efficacy in women with early and advanced BC regardless of menopausal status [11]. On the other hand, the estrogenic effect of tamoxifen in various tissues and organs increases the risk of thromboembolism, endometrial hyperplasia, polyps and cancer, and fatty liver. Furthermore, tamoxifen also reduces the risk of bone loss in post-menopausal women. Based on current evidence it appears that tamoxifen does not cause direct genomic alternations in endometrial cells and rather mediates its carcinogen effect on epithelial cells via estrogenic and non-genomic pathways [8–10,16].

### 3.1.1. Primary Prevention (Chemoprevention)

Both tamoxifen and raloxifene are approved for primary prevention of BC in high-risk women such as age ≥35 years with a history of chest wall radiation prior to 30 years of age or lobular cancer in situ or >1.66% five-year risk of BC as per the modified Gail model [17]. A systemic review of RCTs showed that 5-years of tamoxifen compared to placebo in 28,421 subjects was associated with a 31% reduction in risk of invasive BC (risk ratio [RR], 0.69 [95% CI; 0.59–0.84]) [18]. Likewise, raloxifene compared to placebo was associated with a 56% reduction in risk of invasive BC (RR, 0.44; 0.24–0.80). However, there was no difference in BC mortality or OS. The STAR study, a randomized Phase 3 clinical trial (RCT), compared tamoxifen to raloxifene as primary prevention in postmenopausal women and showed that tamoxifen was slightly more effective at preventing noninvasive BC (RR 1.40; 0.98–2.02), however, compared to raloxifene, it was associated with a higher risk of endometrial cancer, thromboembolic events and cataracts [19] (Table 2). Since primary prevention with tamoxifen has not been associated with a reduction in BC mortality, this intervention requires an open discussion about the potential risk of uterine cancer and thromboembolism.

### 3.1.2. Adjuvant Therapy
#### Ductal Carcinoma In Situ (DCIS)

About 60–75% of all DCIS are ER+ [20,21]. Tamoxifen has been evaluated for secondary prevention of DCIS or invasive cancer in women with DCIS following breast-conserving surgery (BCS) with or without adjuvant radiation therapy [22–25]. The benefit of adjuvant tamoxifen is mostly confined to ER + DCIS [26]. A meta-analysis of 2 RCTs involving 3375 women has shown that 5 years of adjuvant tamoxifen following BCS has been associated with 25% relative reduction in ipsilateral DCIS (HR 0.75; 0.61–0.92) and 21% reduction in ipsilateral invasive BC (HR 0.79; 0.62–1.01); and 50% reduction in contralateral DCIS (RR 0.50; 0.28–0.87) and 43% reduction in contralateral invasive BC (RR 0.57; 0.39–0.83) [25]. The standard dose of tamoxifen is 20 mg for a five-year duration. A recent RCT compared 5 mg of tamoxifen for three years to placebo and showed a 50% relative reduction in BC event (HR, 0.48; 0.26–0.92) [27]. However, there is no overall survival benefit of adjuvant endocrine therapy in DCIS following adjuvant radiation treatment.

#### Invasive Breast Cancer

Five years of adjuvant treatment with tamoxifen in women with early-stage HR + BC has been associated with a significant reduction in recurrence and BC mortality [11,12]. A meta-analysis of RCTs showed that five years of adjuvant tamoxifen in 10,645 women with HR + BC was associated with 47% (RR, 0.53) and 32% (RR, 0.68) reduction in recurrences in the first four years and years 5–10, respectively [11]. The benefit of tamoxifen was independent of age, nodal status, PR status, and adjuvant chemotherapy. There were 29% (RR, 0.71), 34% (RR, 0.66), and 32% (RR, 0.68) significant reduction in breast cancer mortality rates during years 0–4, years 5–9, and years 10–14; respectively. Five to ten years of tamoxifen alone or in combination with ovarian suppression is one of the standard adjuvant endocrine treatments in premenopausal women with HR + BC. Recent evidence has shown that gene expression profile predicts the benefit of adjuvant chemotherapy and using this tool, chemotherapy can be avoided in many women with HR + BC [28,29].

#### Extended Tamoxifen

Results from two large RCTs, ATLAS, and aTTom in women treated with 5-years of adjuvant tamoxifen showed that extended treatment was significantly associated with better outcomes [30,31]. In the ATLAS trial, 12,894 women were randomized to extended tamoxifen versus placebo. During years 5–14, the cumulative risk of recurrence with extended tamoxifen was 21.4% versus 25.1% with placebo. Likewise, during years 5–14, breast cancer mortality with extended tamoxifen was 12.2% compared to 15.0% with observation alone [30]. In the aTTom trial that involved 6953 women, extended tamoxifen

was associated with significantly low recurrence (16.7% vs. 19.3%) and BC mortality (24.5% vs. 26.1%) [31]. The risk of endometrial cancer with extended tamoxifen was 2.2 (1.31–2.34, $p < 0.0001$). Based on the results of these two large trials, extended tamoxifen is a valid option in women with HR + BC. Given the fact that it has been associated with an increased risk of endometrial cancer, it is important to consider risk and benefit for each individual.

The efficacy of tamoxifen in neoadjuvant and metastatic settings are discussed in the aromatase inhibitor section below.

### 3.2. Aromatase Inhibitors

AIs inactivate the aromatase enzyme that converts androgens to estrogens and thereby suppress plasma estrogen level. Currently, three AIs are available; anastrozole and letrozole are two nonsteroidal AIs, whereas exemestane is a steroidal AI [13]. Adjuvant AIs in postmenopausal women with early stage breast cancer have been compared to tamoxifen, tamoxifen in sequence with an AI, combination of tamoxifen and AI or a different AI.

### 3.2.1. Primary Prevention

Anastrozole and exemestane have been evaluated for BC prevention in high-risk postmenopausal women [32,33]. The long-term follow-up data showed that 5 years of anastrozole as primary prevention was associated with a 49% reduction in BC compared to placebo (HR, 0.51; 0.39–0.66) [32]. In a much larger study, 5 years of exemestane was associated with a 65% relative reduction in the annual incidence of invasive BC (HR, 0.35; 0.18–0.70) [33]. Again, similar to SERMS, no survival benefit has been noted.

### 3.2.2. Adjuvant Therapy
DCIS

The NSABP B-35 trial randomized 3104 postmenopausal women with HR + DCIS to 5 years of anastrozole or tamoxifen following lumpectomy and adjuvant radiation therapy [34]. Adjuvant anastrozole compared to tamoxifen, was associated with a 27% relative reduction in DCIS or invasive BC (HR 0.7; 0.56–0.96). Ten-year estimated BC-free survival was 93.1% with anastrozole versus 89.1% with tamoxifen, with 92% OS in both groups. However, anastrozole was superior only in women younger than 60 years of age.

Early-Stage Invasive Cancer

Adjuvant AI vs. Tamoxifen

Several RCTs compared the efficacy of AI to tamoxifen in HR + BC [35,36]. A meta-analysis of individual data of 31,920 postmenopausal women with HR + BC evaluated the efficacy of 5 years of AI to tamoxifen or sequential tamoxifen and AI for a total of 5 years [13]. Overall, adjuvant AI compared to tamoxifen was associated with a significant reduction in risk of recurrence (recurrence rate ratio (RR) 0.70) and BC mortality (RR 0.79; 0.67–0.92). However, the risk reduction was significant only during periods when treatments differed between the two groups. In addition, overall mortality was significantly lower in the AI group (RR 0.88; 0.82–0.94). In subgroup analysis, 5 years of AI compared to tamoxifen was associated with significant reduction in risk of recurrence at years 0–1 (RR of 0.64) and years 2–4 (RR 0.80) but was not significant after that. The ten-year BC mortality rate was 12.1% with AI compared to 14.2% with tamoxifen (RR 0.85, 0.75–0.96). Based on these outcomes, AIs are the preferred option in postmenopausal women with HR + BC.

Adjuvant AI vs. Sequential Tamoxifen and AI

In RCTs that compared 5 years of AI to sequential tamoxifen and AI, overall, fewer recurrences were noted with AI (RR 0.90; 0.81–0.99) however, mortality reduction was not significant (RR 0.89, 0.78–1.03) [13]. Recurrences were significantly low only during years 0–1 (RR of 0.74) while taking tamoxifen, and no significant difference in the rate of recurrences was noted when both groups were taking AI. Sequential AI and tamoxifen were however, more effective than five years of tamoxifen in reducing recurrence during years 2–4 (RR 0.56, 0.46–0.67) but not later. Furthermore, 10-year mortality was 8.7% with

sequential tamoxifen and AI compared to 10.1% with tamoxifen alone ($p = 0.01$). Ten-year incidence of endometrial cancer was 0.4% with AI compared to 1.2% with tamoxifen (RR 0.33; 0.21–0.51), while the 5-year rate of bone fracture was 8.2% with AI compared to 5.5% with tamoxifen (RR 1.42; 1.28–1.57).

Adjuvant Non-Steroidal AI vs. Steroidal AI

At least three RCTs compared different AIs in early-stage HR + BC and showed that they are similar in efficacy and tolerability [37–39]. For example, the MA27 trial compared the efficacy of adjuvant exemestane to anastrozole in 7576 postmenopausal women [39]. At four years, the event-free survival rates were 91% for exemestane and 91.2% for anastrozole. Overall, vasomotor and musculoskeletal symptoms were similar but the risk of bone loss, vaginal bleeding, and abnormal lipid profile were less frequent with exemestane, whereas mild liver function abnormalities and rare episodes of atrial fibrillation were less frequent with anastrozole. However, tolerance to specific AI varies from person to person, and one may tolerate an AI better than the other [37,39]. Therefore, in the setting of poor tolerance to an AI, switching to another AI is an appropriate option.

Adjuvant AI in Premenopausal Women

About one-third of women with BC are premenopausal [40]. There is evidence that in a selected group of premenopausal women, adjuvant AI in combination with ovarian suppression is more effective than adjuvant tamoxifen alone. Ovarian function can be suppressed by luteinizing-hormone-releasing hormone agonists, surgical removal of ovaries, or rarely, irradiation. Results from TEXT and SOFT trials suggest that ovarian suppression in combination with exemestane or tamoxifen is superior to tamoxifen alone in premenopausal women, who are at high risk for recurrence or ≤35 years of age [41–43]. Updated results showed that 8-year disease-free survival (DFS) with tamoxifen alone was 78.9%, but women who received a combination of tamoxifen or exemestane and ovarian suppression had DFS of 83.2% and 85.9%, respectively [44]. However, combination therapy has been associated with high rates of toxicities. For example, hot flushes, sweating, decreased libido, vaginal dryness, insomnia, depression, musculoskeletal symptoms, hypertension, and glucose intolerance are reported more frequently with combination therapy compared to tamoxifen.

Adjuvant Extended Aromatase Inhibitor

Similar to extended tamoxifen, several RCTs have evaluated the benefit of extended AI. A meta-analysis of 12 RCTs involving 24,912 women assessed the benefit of extended 3–5 years AI versus observation after completion of ≥5 years of endocrine therapy [45]. Overall, extended AI was associated with 24% reduction in the risk of recurrence (9.5% vs. 7.0%; $p < 0.00001$), 15% reduction in the risk of distant recurrence (6.1% vs. 5.1%; $p = 0.004$) and nonsignificant reduction in breast cancer-related death (3.1% vs. 2.8%; $p = 0.09$). The benefit of extended AI varied according to the type of prior endocrine therapy. Women who received 5 years of tamoxifen experienced a 33% reduction in the risk of recurrence with a 5-year gain of 3.6% ($p < 0.0001$) as compared to a 24% reduction in the risk of recurrence in women who received 5 years of an AI with a 5-year gain of 1.2% ($p = 0.02$). For women with larger tumors or node-positive disease, who continue to be at high risk of recurrence, extended AIs are an appropriate option.

**Table 2.** Summary of key Phase III clinical trials and systemic review/meta-analysis that evaluated the benefit of antiestrogen therapy including serum estrogen receptor modulators or an aromatase inhibitor (5 years or extended treatment) in primary and secondary prevention of breast cancer.

| Study | Patient Population | Intervention | Outcomes |
|---|---|---|---|
| Nelson et al. (systemic review of 4 trials) [18] | 28,421 women with an increased risk of primary breast cancer | Tamoxifen for 5 years vs. placebo | Tamoxifen compared to placebo resulted in low risk of invasive breast cancer, RR, 0.69 (0.59–0.84) |
| STAR [19] | 19,747 postmenopausal women with an increased risk of primary breast cancer | Tamoxifen or raloxifene for 5 years for primary prevention | Invasive breast cancer, 4.3 per 1000 with tamoxifen vs. 4.41 per 1000 with raloxifene RR 1.02 (0.82–1.28, $p = 0.96$) |
| Staley et al. (systemic review of 2 trials) [25] | 3375 women with DCIS | 5 years of tamoxifen or placebo ± adjuvant radiation therapy | Tamoxifen resulted in 25% reduction in ipsilateral DCIS (HR 0.75; 0.61–0.92); 21% reduction in ipsilateral invasive BC (HR 0.79; 0.62–1.01); 50% reduction in contralateral DCIS (RR 0.50; 0.28–0.87); 43% reduction in contralateral invasive BC (RR 0.57; 0.39–0.83) |
| EBCTCG meta-analysis [11] | 10,645 women with early stage HR+ breast cancer | 5 years of tamoxifen vs. placebo or observation | Adjuvant tamoxifen resulted in 47% (RR, 0.53) and 32% (RR, 0.68) reduction in recurrences in the first four years and year 5–10, respectively. |
| BIG 1-98 [35] | 8010 postmenopausal women with HR+ early breast cancer | 5 years of tamoxifen or letrozole or sequential treatment with two years of one of these agents followed by three years of the other | Letrozole was better than tamoxifen, with 8 year-DFS HR 0.82 (95% CI 0.74–0.92), OS HR 0.79 (0.69–0.90). Eight-year DFS and OS for letrozole, letrozole followed by tamoxifen, and tamoxifen followed by letrozole were 78.6%, 77.8%, 77.3% and 87.5%, 87.7%, 85.9% respectively |
| NCIC CTG MA27 [39] | 7576 postmenopausal women with HR+ early breast cancer | 5 years of exemestane versus anastrozole | 4-year EFS was 91% for exemestane and 91.2% for anastrozole (HR,1.02; 95% CI, 0.87 to 1.18, $p = 0.85$) |
| ATLAS trial [30] | 12,894 women with HR+ early stage breast cancer completed 5 years of tamoxifen | Extended 5 years of tamoxifen vs. observation after completion of 5 years of tamoxifen | During years 5–14, the cumulative risk of recurrence with extended tamoxifen was 21.4% versus 25.1% with observation, $p = 0.002$; breast cancer mortality 12.2% with tamoxifen vs. 15.0% with observation, $p = 0.01$ |
| Gary et al. (EBCTCG meta-analysis) [45] | 24,912 postmenopausal women with HR+ early breast cancer | Extended 3–5 years aromatase inhibitor vs. observation or placebo after completion of ≥5 years of adjuvant endocrine therapy | Extended AI was associated with 24% reduction in the risk of recurrence (9.5% vs. 7.0%; $p < 0.00001$), 15% reduction in the risk of distant recurrence (6.1% vs. 5.1%; $p = 0.004$) and nonsignificant reduction in breast cancer mortality (3.1% vs. 2.8%; $p = 0.09$) |
| SOFT and TEXT [42] | 4690 premenopausal women with HR+ early breast cancer | Exemestane plus ovarian suppression or tamoxifen plus ovarian suppression for 5 years | 5 years DFS 91.1% with exemestane–ovarian suppression vs. 87.3% with tamoxifen–ovarian suppression, HR: 0.72 (0.60–0.85) |

AI: aromatase inhibitor; BC: breast cancer; DCIS: ductal carcinoma in situ; DFS: disease free survival; EBCTCG: Early Breast Cancer Trialists' Collaborative Group; EFS: event free survival; HR+: hormone receptor positive; OS: overall survival; RR: risk ratio.

### 3.2.3. Neoadjuvant AI

A meta-analysis of 20 RCTs involving 3490 subjects evaluated the efficacy of AIs to chemotherapy or tamoxifen [46]. Compared with combination chemotherapy, AI monotherapy had a similar clinical response rate, radiological response rate, and breast-conserving surgery (BCS) rate with lower adverse effects. However, AIs compared with tamoxifen were associated with a significantly higher clinical response rate, radiological response rate, and BCS rate [46]. Therefore, neoadjuvant AIs are a preferred option in women with locally advanced HR + BC if chemotherapy is relatively contraindicated.

### 3.2.4. Advanced Breast Cancer

For most individuals with HR + BC, endocrine therapy in combination with a targeted agent is the preferred option (discussed below). AI monotherapy is an effective first-line option in HR + mBC. A meta-analysis of 25 RCTs involving 8504 subjects showed that AIs were more effective than tamoxifen [14]. For example, in the first-line setting, AIs compared to tamoxifen were associated with an 11% (HR 0.89; 0.80–0.99) reduction in mortality. Likewise, compared to other treatments, AIs were associated with a 14% (HR 0.87: 0.82–0.93) reduction in mortality in second- and subsequent-line trials. Evidence does not support one AI over another based on its efficacy [47,48]. The data, however, supports that following progression, switching from a nonsteroidal to a steroidal AI or vice versa is an effective approach in HR + mBC [49].

### 3.3. Selective Estrogen Receptor Degraders (SERDs): Fulvestrant

Fulvestrant is the only SERD that is currently used in clinical practice [50]. It is a pure ER antagonist, exerting selective ER downregulation, and competitively binding to the ER [51,52]. It is administered as an intramuscular injection. A Phase III trial examined the optimal dose of fulvestrant by randomly assigning 736 postmenopausal women with HR + mBC to fulvestrant 250 mg vs. 500 mg dose. Progression-free survival (PFS), the primary endpoint of the study, was better with fulvestrant 500 mg (HR 0.80; 0.68–0.94) [53]. Follow up of the study showed superior mOS with fulvestrant 500 mg, 26.4 vs. 22.3 months (HR, 0.81; 0.69–0.96) [54].

### 3.3.1. Fulvestrant vs. Aromatase Inhibitor

The efficacy of fulvestrant compared with anastrozole has been demonstrated in an RCT involving 462 women with HR + mBC, all of whom were ET naïve [15]. Women who received fulvestrant had mPFS of 16.6 vs. 13.8 months if they received anastrozole (HR, 0.80; 0.64–0.99). Fulvestrant and exemestane are equally active and well-tolerated in women with mBC who have experienced progression or recurrence during treatment with a nonsteroidal AI [55]. Fulvestrant in combination with targeted therapy is more effective than monotherapy in the second-line setting (discussed below).

### 3.3.2. Fulvestrant in Combination with AI

Several trials have evaluated the combination of fulvestrant and an AI in first-line therapy for mBC [56–58]. Although an RCT involving 694 postmenopausal women with HR + mBC showed better PFS (15 months vs. 13.5 months, HR 0.80; 0.68–0.94) and OS (47.7 months vs. 41.3 months; HR 0.81; 0.65–1.00) with a combination of fulvestrant and anastrozole compared to anastrozole [56], two other RCTs failed to demonstrate the benefit of a combination of fulvestrant and AI [57,58]. It is important to note that in this trial, almost 40% of women had metastatic disease at presentation. In a subgroup analysis, the survival difference was significant in women who did not receive prior tamoxifen therapy (HR, 0.74; 95% CI: 0.56 to 0.98) but not among women who had received prior tamoxifen therapy (HR, 0.91; 95% CI: 0.65 to 1.28). A combination of fulvestrant and targeted therapy is reviewed below.

## 4. Cyclin-Dependent Kinase (CDK) 4/6 Inhibitors

Three orally bioavailable CDK 4/6 inhibitors, palbociclib, ribociclib, and abemaciclib are currently approved for use in HR + mBC (Table 1). CDK4 and CDK6 are enzymes responsible for phosphorylating retinoblastoma protein to inactivate it. Inactivated retinoblastoma protein releases transcription factors that promote cell cycle progression from the $G_1$ phase to the S phase [59]. Therefore, CDK 4/6 inhibitors initiate $G_1$ arrest by blocking the phosphorylation and inactivation of retinoblastoma protein [60]. The retinoblastoma protein is commonly intact in BC, making CDK 4/6 inhibition a viable therapeutic option [61].

### 4.1. CDK 4/6 Inhibitors in Combination with an Aromatase Inhibitor (First-Line)

CDK 4/6 Inhibitors are approved as an adjunct treatment to AI in patients with HR + mBC. Several RCTs have confirmed that the addition of CDK 4/6 inhibitors to an AI in a first-line setting has been associated with almost doubling of mPFS (~10 months) with an acceptable toxicity profile [6] (Table 3).

Following the PALOMA-1/TRIO-18 results, a Phase II study that demonstrated increased mPFS of 20.2 months with palbociclib and letrozole vs. 10.2 months with letrozole, palbociclib received accelerated approval for use in combination with letrozole as initial therapy in postmenopausal women [62]. The survival benefit between the two arms was not significant; 37.5 months in the palbociclib group vs. 34.5 months in the placebo group (HR 0.897; 0.62–1.29) [63]. PALOMA-2 was a Phase III study that randomized 666 treatment naïve postmenopausal women with HR + mBC at 2:1 to letrozole plus palbociclib versus letrozole and placebo and confirmed the benefit of combination therapy with an mPFS of 24.8 months with palbociclib vs. 14.5 months with placebo (HR 0.58; 0.46–0.72) [64].

Subsequently, ribociclib and abemaciclib also received approval for their use in combination with an AI for HR + mBC following the results of MONALEESA-2 and MONARCH-3, respectively [65,66] (Table 3). In the MONALEESA-2 trial, 668 postmenopausal women were randomized to letrozole plus ribociclib or placebo. The group with ribociclib plus letrozole had a significantly superior mPFS of 25.3 months compared to 16 months with letrozole alone (HR 0.57; 0.46–0.70) [65]. In the MONARCH-3 trial, 493 postmenopausal women were randomized to abemaciclib or placebo plus a nonsteroidal AI [66,67]. Abemaciclib was associated with a significantly prolonged mPFS of 28.18 months vs. 14.76 months with placebo (HR, 0.54; 0.42–0.70) [67]. Ribociclib was also approved for use in pre/perimenopausal women following the results of MONALEESA-7. In the MONALEESA-7 trial, 672 women were randomized to ribociclib or placebo in combination with goserelin and either a nonsteroidal AI or tamoxifen [68]. Unlike other first-line trials, women who had received one previous line of chemotherapy for advanced disease were permitted. The mPFS of the ribociclib group was 23.8 months compared to 13.0 months with placebo (HR, 0.55; 0.44–0.69) [69]. The improvement in mOS at 42 months was significant in MONALEESA-7, with 70.2% surviving in the ribociclib group compared to 46.0% in the placebo group (HR 0.71, 95% CI 0.54–0.95) [68]. OS data for the other studies evaluating first-line CDK 4/6 inhibitors are pending.

Improvements in mPFS when added to an AI are similar in each CDK 4/6 inhibitor (~10 months), with hazard ratios ranging from 0.54 to 0.58 across the Phase III trials [64–66]. Furthermore, the results of MONALEESA-7 showed that the addition of CDK 4/6 inhibitors in pre/perimenopausal women with endocrine therapy resulted in a similar benefit and safety profile that is seen in postmenopausal women [69]. MONALEESA-7 also demonstrated that adding CDK 4/6 inhibitors to tamoxifen results in similar improvement in PFS seen with AIs [61,68,69]. However, a combination of ribociclib and tamoxifen may further increase the risk of Q-T prolongation. It is important to note that a randomized Phase II trial has demonstrated better PFS with a combination of CDK 4/6 inhibitor and an AI vs. single-agent capecitabine in HR + mBC in the first-line setting [70].

**Table 3.** Clinical data of CDK 4/6 inhibitors with an aromatase inhibitor as first line treatment for women with HR+ advanced breast cancer.

| Parameters | PALOMA-1/TRIO-18 [62,63] | PALOMA-2 [64] | MONALEESA-2 [65] | MONALEESA-7 [68,69] | MONARCH-3 [66,67] |
|---|---|---|---|---|---|
| Patient population | Postmenopausal $n = 165$ | Postmenopausal [a] $n = 666$ | Postmenopausal [a] $n = 668$ | Pre/perimenopausal $n = 672$ | Postmenopausal [a] $n = 493$ |
| Treatment arms | Palbociclib vs. no palbociclib | Palbociclib vs. placebo | Ribociclib vs. placebo | Ribociclib vs. placebo | Abemaciclib vs. placebo |
| Hormonal therapy | Letrozole | Letrozole | Letrozole | Goserelin + ET [b] | NSAI [c] |
| ORR [d] | 55% vs. 39% | 55.3% vs. 44.4% | 54.5% vs. 38.8% | 51% vs. 36% | 61.0% vs. 45.5% |
| Median PFS (months) HR (95% CI) | 20.2 vs. 10.2 0.488 (0.319–0.748; $p = 0.0004$) | 24.8 vs. 14.5 0.58 (0.46–0.72, $p < 0.001$) | 25.3 vs. 16.0 0.568 (0.457–0.704, $p < 0.001$) | 23.8 vs. 13.0 0.55 (0.44–0.69, $p < 0.001$) | 28.18 vs. 14.76 0.540 (0.418–0.698, $p < 0.001$) |
| Median OS (months) HR (95% CI) | 37.5 vs. 34.5 0.897 (0.623–1.294, $p = 0.28$) | Pending | Pending | At 42 months: 70.2% vs. 46.0% 0.71 (0.54–0.95, $p = 0.0097$) | Pending |

[a] Prior adjuvant or neoadjuvant treatment with a nonsteroidal aromatase inhibitor was allowed unless disease had recurred while the patient was receiving the therapy or within 12 months after completing therapy. [b] either tamoxifen or an NSAI; [c] either anastrozole or letrozole; [d] in those with measurable disease ORR: objective response rate, PFS: progression free survival, OS: overall survival, HR: hazard ratio, CI: confidence interval, ET: estrogen therapy, NSAI: nonsteroidal aromatase inhibitor.

Adverse effects related to CDK 4/6 inhibitors are leukopenia (less common with abemaciclib), fatigue, nausea, diarrhea (more common with abemaciclib), liver dysfunction, and anemia [64–66]. A rare but notable adverse effect of ribociclib is QTcF interval prolongation, which occurred in 3.3% patients, mostly within the first 4 weeks of treatment [65,69]. Based on the available data, combination of AI and a CDK 4/6 inhibitor is the standard first-line treatment option for most individuals with HR + mBC.

### 4.2. CDK 4/6 Inhibitors in Combination with Fulvestrant (First or Subsequent Line)

The addition of CDK 4/6 inhibitors in patients taking fulvestrant is recommended in patients who previously progressed on endocrine therapy. This recommendation is based on improvements in mPFS (6–7 months) and quality of life [6]. Clinical data of the three CDK 4/6 inhibitors in combination with fulvestrant has been demonstrated by three major RCTs (Table 4).

**Table 4.** Clinical data of CDK 4/6 inhibitors with fulvestrant in women with HR+ advanced breast cancer who progressed on adjuvant or 1st line endocrine therapy.

| Parameters | PALOMA-3 [71,72] | MONALEESA-3 [73,74] | MONARCH-2 [75,76] |
|---|---|---|---|
| Patient population | Any menopausal status Up to one line of prior chemotherapy in advanced setting $n = 521$ | Postmenopausal No prior chemotherapy in advanced setting $n = 726$ | Any menopausal status No prior chemotherapy in advanced setting $n = 669$ |
| Line of endocrine therapy | Second or later line | First or second line | Second line |
| Treatment arms | Palbociclib vs. placebo | Ribociclib vs. placebo | Abemaciclib vs. placebo |
| Hormonal therapy | Fulvestrant +/− LHRH agonist [a] | Fulvestrant | Fulvestrant +/− LHRH agonist [a] |
| ORR | 24.6% vs. 10.9% | 40.9% vs. 28.7% | 48.1% vs. 21.3% |
| Median PFS (months) HR (95% CI) | 9.5 vs. 4.6 0.46 (0.36–0.59, $p < 0.001$) | 20.5 vs. 12.8 0.593 (0.480–0.732, $p < 0.001$) | 16.4 vs. 9.3 0.553 (0.449–0.681, $p < 0.001$) |
| Median OS (months) HR (95% CI) | 34.9 vs. 28.0 0.81 (0.64–1.03, $p = 0.09$) | Not reached vs. 40 0.72 (0.57–0.92, $p = 0.0045$) | 46.7 vs. 37.3 0.757 (0.606–0.945, $p = 0.01$) |

[a] In pre/perimenopausal women; ORR: objective response rate, PFS: progression free survival, OS: overall survival, HR: hazard ratio, CI: confidence interval, LHRH: luteinizing hormone-releasing hormone.

Following the results of PALOMA-3, palbociclib was approved in combination with fulvestrant for women that developed disease progression during endocrine therapy. PALOMA-3 studied 521 women of any menopausal status who had disease progression after previous endocrine therapy in the advanced setting or within 12 months of completing adjuvant therapy. The mPFS was 9.5 months in patients given fulvestrant plus palbociclib vs. 4.6 months with fulvestrant alone (HR 0.46; 0.36–0.59) [71]. The mPFS in patients given palbociclib was similar between pre/perimenopausal women (9.5 months, HR 0.50; 0.29–0.87) and postmenopausal women (9.9 months, HR; 0.45; 0.34–0.59) [71]. The difference in mOS was not significant, at 34.9 months in the palbociclib group vs. 28.0 months in the placebo group (HR 0.81; 0.64–1.03) [72]. However, women who had an endocrine sensitive disease and received palbociclib had a significant better survival with mOS of 39.7 months compared to 29.7 months in women with placebo (HR 0.72; 0.55–0.94) [72].

After the MONALEESA-3 study, ribociclib was approved in combination with fulvestrant for postmenopausal women as initial endocrine-based therapy or following disease progression on endocrine therapy. Postmenopausal women included in MONALEESA-3 were treatment naïve, or had received up to one line of prior endocrine therapy [73]. The mPFS was 20.5 months in the ribociclib group vs. 12.8 in the placebo group (HR 0.593; 0.480–0.732). Improvements in PFS were consistent between patients that were treatment naïve (HR 0.58; 0.42–0.80) or those that had received one line of prior endocrine therapy (HR 0.57; 0.43–0.74) [73]. The increase in OS at 42 months on ribociclib vs. placebo was significant, estimated as 57.8% vs. 45.9% (HR 0.72; 0.57–0.92) [74].

Following MONARCH-2, abemaciclib was approved in combination with fulvestrant in patients with disease progression following endocrine therapy. MONARCH-2 was a study with women of any postmenopausal status that had disease progression after previous endocrine therapy or within 12 months of completing adjuvant therapy. Patients that received abemaciclib had significant improvements in mPFS, of 16.4 months vs. 9.3 months with placebo (HR 0.553; 0.45–0.68) [75]. The improvement in mOS in abemaciclib vs. placebo was also significant, 46.7 months vs. 37.3 months (HR 0.76; 0.61–0.95) [76]. Taken together, a combination of fulvestrant and a CDK 4/6 inhibitor is a standard option for women with HR + mBC who have progressed on first-line AIs or tamoxifen.

### 4.3. Abemaciclib as Monotherapy (Subsequent Line)

Abemaciclib is the only CKD 4/6 inhibitor approved for monotherapy, following the results from the MONARCH-1 trial in women who progressed on prior endocrine therapy and had 1 or 2 chemotherapy regimens. The confirmed objective response rate (ORR) was 19.7% at 12 months [77]. In this setting, chemotherapy would have an expected ORR of 10–20%, which was on par to the ORR of abemaciclib. The mPFS was 6.0 months, and the mOS was 17.7 months [77].

### 4.4. Adjuvant CDK 4/6 Inhibitors

MONARCH-E evaluated the efficacy of adjuvant abemaciclib in 5637 subjects with high-risk early-stage HR + BC [78,79]. Individuals with ≥4 positive nodes, or 1–3 nodes and either tumor size ≥5 cm, grade 3, or Ki-67 ≥ 20%, were randomized to 2-years of adjuvant abemaciclib with endocrine therapy or endocrine therapy alone. An interim analysis when 25.5% of women completed 2-year of adjuvant abemaciclib showed that abemaciclib in combination with endocrine therapy was associated with a 29% reduction in invasive breast cancer recurrence (HR, 0.71; 0.58–0.87) with 2-year invasive DFS of 92.3% compared to 89.3% with endocrine therapy alone. No new safety concerns were observed. However, the data is not mature and further follow up will be important to determine the magnitude of early separation of invasive DFS curves and overall survival benefit.

In contrast, the PALLAS trial that randomized 5760 subjects with Stage 2 and 3 disease to 2-years of palbociclib with adjuvant endocrine therapy or endocrine therapy alone failed to show benefit of adjuvant palbociclib [80]. After a median follow-up of 23.7 months, the invasive DFS rate was 88.2% in palbociclib plus endocrine group compared to 88.5% in

the endocrine alone group. Of note, 42.2% of subjects discontinued palbociclib prior to the planned 2-year duration, primarily due to adverse events.

Likewise, the PENELOPE-B study randomized 1250 patients who had a less than pathologic complete response following neoadjuvant chemotherapy to adjuvant endocrine therapy plus palbociclib or placebo for 1 year [81]. At a median follow-up period of 43 months, no difference in invasive DFS was noted between the two groups (HR 0.93; 0.74–1.17). The differences in the outcomes of the three trials could be due to a different patient population in each study. For example, MONARCH-E enrolled women with a much higher risk of recurrences. Longer follow-up, including survival data, is required before CDK 4/6 inhibitors could be considered in the adjuvant setting. Furthermore, biomarkers analysis of tumor tissues from all three trials may help in identifying a subset of women who benefit from CDK4/6 inhibitors.

## 5. PI3K/AKT/mTOR Pathway Inhibitors

The PI3K/Akt/mTOR (PAM) pathway is frequently aberrant in BC, and activation of this pathway is associated with resistance to endocrine therapy [82]. The PAM pathway is involved in regulating the cell cycle. In the first step of the pathway, phosphatidylinositol 3-kinases (PI3K) are activated after growth factor or ligand binding. Through a series of phosphorylation steps, PI3K activation leads to the activation of Akt, which activates mTORC1 [82]. mTORC1 then stimulates cell growth by activating translation factors [83]. There is a good rationale for targeting the key mediators of this pathway, P13K, Akt, and mTORC1. Furthermore, PIK3CA, a class 1A PI3K, is mutated and hyperactive in about 40% of patients with HR + mBC [84]. Two inhibitors of the PAM pathway are currently approved in HR + mBC, everolimus (an mTOR inhibitor), and alpelisib (a PI3KCA inhibitor) (Tables 1 and 5).

### *5.1. mTOR Inhibitors*

5.1.1. Temsirolimus in Combination with Letrozole (First-Line)

The HORIZON trial examined the efficacy of temsirolimus, an mTOR inhibitor, versus placebo, in combination with letrozole in 1112 patients in treatment naive, HR + mBC [85]. There was no improvement in the primary endpoint of PFS (9 months; HR, 0.90; 0.76–1.07). However, an exploratory analysis showed improved mPFS in patients <66 years if they received temsirolimus (9.0 vs. 5.6 months; HR, 0.75; 0.60 to 0.93).

5.1.2. Everolimus in Combination with Exemestane (Subsequent Line)

In BOLERO-2, 724 postmenopausal women who were randomized to everolimus (an oral mTOR inhibitor) and exemestane at 2:1 had a median PFS of 7.8 months vs. 3.2 months with exemestane alone (HR 0.45; 0.38–0.54) [86]. The mOS was 31.0 months for women receiving everolimus and exemestane vs. 26.6 months in those receiving only exemestane (HR 0.89; 0.73–1.10) [87]. Although this translated to an absolute increase of 4.4 months, this was not statistically significant. Following the results, everolimus was approved in combination with exemestane for patients with HR + mBC that had disease progression on nonsteroidal AIs. Stomatitis was a common adverse effect of everolimus, occurring in 59% of patients [86]. Other side effects included rash, fatigue, diarrhea, nausea, and decreased appetite. Less common but important side effects were noninfectious pneumonitis (16%), hyperglycemia (14%), and urinary tract infections (10%) [86]. There is a lack of prospective data of the efficacy of everolimus in women who previously received a combination of AI and a CDK 4/6 inhibitor.

### *5.2. PI3KCA Inhibitors*

5.2.1. Buparlisib in Combination with Fulvestrant (Subsequent Line)

Two RCTs evaluated the efficacy of the pan-PI3K inhibitor, buparlisib, with a primary endpoint of PFS. BELLE-2 randomized 1147 patients with HR + mBC who were previously treated with endocrine therapy, to oral buparlisib plus fulvestrant or fulvestrant

alone [88,89]. In unselected patients, mPFS was 6.9 months in the buparlisib group versus 5.0 months in the placebo group, HR 0.78 (0.67–0.89). About 23% of patients experienced serious adverse effect, including liver dysfunction and hyperglycemia. In BELLE-3, 432 postmenopausal women who progressed on endocrine therapy and mTOR inhibitors were randomly assigned buparlisib plus fulvestrant or fulvestrant alone [90]. Median PFS was significantly better in the buparlisib group (3.9 months vs. 1.8 months; HR 0.67; 0.53–0.84), but high rates of adverse events were noted. Due to the modest benefit and toxicity associated with buparlisib, further studies are not planned.

### 5.2.2. Taselisib in Combination with Fulvestrant (Subsequent Line)

The Phase III SANDPIPER trial assessed taselisib, a potent, selective PI3K inhibitor, plus fulvestrant in HR + mBC [91]. PFS was significantly improved in the taselisib group (7.4 months vs. 5.4 months, HR 0.70; 95% CI, 0.56–0.89). AEs included diarrhea, hyperglycemia, and colitis. About 32% experienced serious adverse effects and 17% of patients in the taselisib group discontinued treatment early.

### 5.2.3. Alpelisib in Combination with Fulvestrant (Subsequent Line)

Alpelisib was the first PI3K3CA inhibitor to be approved in combination with fulvestrant for patients with HR + mBC with a PIK3CA mutation who progressed on endocrine therapy [84]. The SOLAR-1 study investigated HR + mBC patients who progressed on endocrine therapy. Patients with a PIK3CA mutation-positive BC who received alpelisib and fulvestrant had a mPFS of 11.0 months vs. 5.7 months with fulvestrant (HR 0.65; 0.50–0.85) [84]. Alpelisib is not recommended for patients that do not have a PIK3CA mutation, as the difference in mPFS in this group was not significant (HR 0.85; 0.58–1.25). With a follow-up of 30.8 months, median OS was 39.3 months with combination therapy vs. 31.4 month with fulvestrant alone (HR 0.86; 0.64–1.15) [91]. Hyperglycemia was noted in 63.7% of patients with alpelisib. Diarrhea, nausea, decreased appetite, rash, and stomatitis were also common [84,92].

**Table 5.** Clinical data of key Phase III trials involving PI3K/Akt/mTOR pathway inhibitors with hormonal therapy as first or subsequent line treatment for patients with HR+ advanced breast cancer.

| Parameters | BOLERO-2 [86,87] | HORIZON [85] | BELLE-2 [88,89] | SOLAR-1 [84,92] |
|---|---|---|---|---|
| Patient population | Postmenopausal women *n* = 724 | Men or postmenopausal women *n* = 1112 | Postmenopausal women *n* = 1147 | Men or postmenopausal women PIK3CA mutation *n* = 572 |
| Line of therapy | Second or later line | First line | Second or later line | Second or later line |
| Treatment arms | Everolimus vs. placebo | Temsirolimus vs. placebo | Buparlisib vs. palcebo | Alpelisib vs. placebo |
| Hormonal therapy | Exemestane | Letrozole | Fulvestrant | Fulvestrant |
| ORR | 12.6% vs. 1.7% | 27 vs. 27% | 11.8 vs. 7.7% | 26.6% vs. 12.8% |
| Median PFS (months) HR (95% CI) | 7.8 vs. 3.2 0.45 (0.38–0.54, *p* = <0.001) | 8.9 vs. 9.0 0.90 (0.76–1.07, *p* = 0.25) | 6.9 vs. 5.0 0.78 (0.67–0.89, *p* = 0.0002) | 11.0 vs. 5.7 0.65 (0.50–0.85, *p* = <0.001) |
| Median OS (months) HR (95% CI) | 31.0 vs. 26.6 0.89 (0.73–1.10, *p* = 0.14) | Not reported (most patients censored) 0.89 (0.65–1.23, *p* = 0.50) | 33.2 vs. 30.4 0.87 (0.74–1.02; 0.045) | 39.3 vs. 31.4 0.86 (0.64–1.15, *p* = 0.15) |

ORR: objective response rate, PFS: progression free survival, OS: overall survival, HR: hazard ratio, CI: confidence interval.

### 6. Histone Deacetylase Inhibitors in Combination with Exemestane

Entinostat is an oral isoform-selective histone deacetylase inhibitor that targets resistance to endocrine treatment in HR + mBC. In a Phase II trial, 130 postmenopausal women with HR + mBC who progressed on a nonsteroidal AI were randomly assigned to exemestane plus entinostat or exemestane plus placebo [93]. Combination treatment resulted in mPFS of 4.3 months vs. 2.3 months with exemestane alone (HR, 0.73; 0.50–1.07). Median OS was 28.1 months vs. 19.8 months (HR, 0.59; 0.36–0.97). Grade 3/4 adverse events included low neutrophil or platelet count, hypophosphatemia, anemia, fatigue, and diarrhea. However, a Phase 3 confirmatory trial that enrolled 608 individuals failed to confirm the benefit of entinostat in combination with exemestane [94]. At final analysis, mPFS was 3.3 months with the combination vs. 3.1 months with exemestane (HR 0.87; 0.67–1.13). The median OS with the combination was 23.4 months vs. 21.7 months (HR 0.99; 0.82–1.21).

A multicenter Chinese Phase III study compared combination of chidamide and exemestane to exemestane and placebo in 365 women in the second-line setting [95]. The trial met its primary endpoint of improvement in PFS of 7.4 months with the combination vs. 3.8 months with exemestane alone (HR, 0.75; 0.58–0.98). Neutropenia, thrombocytopenia, and leukopenia were the most common adverse effects. Further data is needed to identify the role of histone deacetylase inhibitor in HR + mBC.

### 7. Anti-VEGF in Combination with Chemotherapy or Endocrine Therapy

An RCT involving 722 patients with advanced HER2-negative BC were treated with a combination of paclitaxel and bevacizumab or paclitaxel alone [96]. Paclitaxel plus bevacizumab significantly prolonged PFS, the primary endpoint, compared to paclitaxel alone (11.8 vs. 5.9 months; HR, 0.60; $p < 0.001$). The OS was similar in the two groups (26.7 vs. 25.2 months; HR, 0.88; $p = 0.16$). Hypertension, proteinuria, headache, and cerebrovascular ischemia were more frequent in patients who received bevacizumab. The US Food and Drug Administration (FDA) in November 2011 withdrew the metastatic breast cancer indication for bevacizumab.

Subsequently, another Phase III trial evaluated the combination of bevacizumab plus letrozole or fulvestrant versus letrozole or fulvestrant as first-line therapy in 380 postmenopausal patients with HR + mBC [97]. The trial failed to reach its primary endpoint of improvement in PFS. Median PFS was 14.4 months in the endocrine arm and 19.3 months in the endocrine-bevacizumab arm (HR, 0.83; 0.65 to 1.06). Hence, the role of anti-VEGF therapy in HR + BC is currently not well defined.

### 8. Other Compounds

Olaparib is an oral poly (adenosine diphosphate–ribose) polymerase inhibitor that has been compared with single-agent chemotherapy in an RCT involving 302 women with advanced HER2-negative BC with a germline *BRCA* mutation [98]. Median PFS was significantly longer in the olaparib group than in the standard therapy group (7.0 months vs. 4.2 months; HR, 0.58; 0.43–0.80). The ORR was 59.9% in the olaparib group and 28.8% in the standard therapy group. Overall survival difference was not significant (HR, 0.90; 0.63–1.29; $p = 0.57$). Likewise, another RCT evaluated talazoparib, a poly (adenosine diphosphate-ribose) inhibitor, versus single agent chemotherapy of the physician's choice in 431 women with a germline BRCA mutation [99]. Median progression-free survival was 8.6 months with talazoparib versus 5.6 months with standard single agent chemotherapy, HR, 0.54 (0.41 to 0.71). The interim median hazard ratio for death was 0.76 (95% CI, 0.55 to 1.06; $p = 0.11$). The ORR was 62.6% in the talazoparib group vs. 27.2% in the standard-therapy group. Hence, olaparib or talazoparib is a treatment option in women with HR + mBC with a germline *BRCA* mutation.

Although epidermal growth factor receptor inhibitors that have shown efficacy in several solid organ cancers have failed to demonstrate benefit in patients with EGFR-expressing HR + mBC [100], the fibroblast growth factor receptors are another potential

target in women with HR + mBC. Several Phase 2 studies have evaluated their efficacy in advanced BC with modest benefit [101]. Lastly, immunotherapy involving immune checkpoint inhibitors have shown efficacy in triple negative breast cancer. However, their role in HR + BC is not well known [102,103].

## 9. Future Directions

### 9.1. Early Stage Disease

Several Phase III trials are evaluating adjuvant CDK 4/6 inhibitors. The NATALEE trial is evaluating the efficacy of ribociclib in Stages 2 and 3 BC (NCT03701334), the WSG ADAPT umbrella trial aims to establish early predictive molecular surrogate markers for a response after a short 3-week induction treatment of ribociclib plus endocrine therapy versus chemotherapy (NCT01779206), and the POLAR study is assessing the benefit of 3 years of palbociclib plus endocrine therapy in patients with isolated locoregional recurrence (NCT03820830). Results of these trials will be helpful to further define the role of adjuvant CDK 4/6 inhibitors.

### 9.2. Metastatic Disease

AMBRE is a Phase III French study comparing chemotherapy to abemaciclib plus endocrine therapy in patients with visceral metastasis and high burden disease (NCT04158362). The RIBBIT study will examine ribociclib plus an AI or fulvestrant to capecitabine with bevacizumab or paclitaxel with/without bevacizumab in patients with advanced BC with visceral metastasis (NCT03462251). A Chinese Phase III clinical is evaluating the efficacy of SHR6390, a novel CDK 4/6 inhibitor in combination with letrozole or anastrozole (NCT03966898).

Somatic mutations in the ER gene ligand-binding domain region results in ligand-independent activated ER, and resistance to anti-estrogen therapy [104,105]. Evidence suggests that fulvestrant may be more effective than AI in *ESR1*-mutated patients [106,107]. Several studies are evaluating novel SERDs in the management of advanced BC. A Phase III study is evaluating the efficacy of an oral SERD, GDC-9545, plus palbociclib vs. letrozole plus palbociclib (NCT04546009). AMEERA-5, a Phase III study, is examining SAR439859, an oral SERD, plus palbociclib to letrozole plus palbociclib in the first-line setting (NCT04478266). Other studies are examining the efficacy of novel AKT/PIK3 pathway inhibitors. A Phase III study is assessing the efficacy of capivasertib, a potent pan-AKT inhibitor with fulvestrant vs. fulvestrant in HR + mBC following recurrence or progression on AI therapy (NCT04305496). Another Phase III trial is evaluating the efficacy of ipatasertib, a drug that binds to all three isoforms of AKT, in combination with palbociclib plus fulvestrant compared to palbociclib plus fulvestrant (NCT04060862). One Phase III trial will assess the efficacy of GDC-0077 in combination with palbociclib and fulvestrant compared to placebo plus palbociclib and fulvestrant in patients with PIK3CA-mutant HR + BC (NCT04191499).

Several Phase 2 trials are examining different strategies to overcome endocrine resistance in HR + mBC. The Phase 1/2 TRINITI-1 trial (NCT02732119) is exploring if adding the mTOR inhibitor, everolimus, to exemestane and ribociclib can overcome resistance to endocrine therapy and CDK4/6 inhibition. Another Phase 2 trial, TBCRC 041 (NCT02860000), is examining alisertib, an oral Aurora A kinase inhibitor with or without fulvestrant in endocrine resistant disease. Several Phase 2 trials are evaluating the efficacy of various fibroblast growth factor receptor (FGFR) inhibitors in FGFR-amplified HR+ disease, including erdafitinib in combination with fulvestrant and palbociclib (NCT03238196), debio 1347 and fulvestrant in (NCT03344536) and TAS-120 (NCT04024436).

Two Phase III studies are comparing sacituzumab govitecan, a Trop-2-directed antibody and topoisomerase inhibitor SN-38 drug conjugate to single agent chemotherapy in HR + BC after the failure of at least 2, and no more than 4, prior chemotherapy regimens (NCT04639986 and NCT03901339). The Phase 2 PACE trial (NCT03147287) is a

3-arm trial that is evaluating synergistic effect of immune checkpoint inhibitor avelumab to combination of fulvestrant and palbociclib.

## 10. Conclusions

HR + BC is a distinct entity. Endocrine therapy remains the backbone treatment for primary and secondary prevention of HR+ early-stage BC. Current available endocrine therapy includes SERMs such as tamoxifen, aromatase inhibitors including anastrozole, letrozole, and exemestane, and SERDs such as fulvestrant. In early-stage disease, tamoxifen or AIs are standard endocrine therapy. In pre-menopausal women who are ≤35 years or those with high-risk disease, combination of tamoxifen or an AI with ovarian suppression is more effective than tamoxifen alone, however, this option has been associated with a high rate of adverse effects. In postmenopausal women, AIs are more effective than tamoxifen and are the preferred option. Recent data has shown conflicting results of adjuvant CDK 4/6 inhibitors in combination with an AI in women with high-risk breast cancer. Endocrine therapy is the key initial treatment for most patients with HR + mBC. However, resistance to antiestrogen treatment is a major barrier in long-term disease control and to evade chemotherapy. Over the past decade, several novel targeted agents combined with antiestrogen therapy, such as CDK 4/6 inhibitors, mTOR/PIK3 inhibitors, and histone deacetylase inhibitors have shown better efficacy with the potential of overcoming endocrine resistance. Combination of an AI or fulvestrant and a CDK 4/6 inhibitor is the current initial standard treatment for individuals with HR + mBC. On progression, an alternate endocrine agent in combination with another targeted agent is a valuable option. The optimal sequence of various targeted agents is not known and there is a need to identify mechanisms of resistance to current established therapies. However, with a better understanding of the biologic pathways and identification of new biomarkers, the treatment landscape of HR + BC is rapidly evolving.

**Funding:** This research received no external funding.

**Conflicts of Interest:** The authors declare no conflict of interest.

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
