# Peer review of "Current Landscape of Targeted Therapy in Hormone Receptor-Positive and HER2-Negative Breast Cancer"

_curroncol, doi:10.3390/curroncol28030168_

Round 1

Reviewer 1 Report

In this review article, the authors thoroughly described the available targeted therapy approaches in the clinic to deal with the Hormone Receptor-Positive and HER2-negative breast cancer (HR+BC), the most notorious subtype of breast cancer. Here is the minor recommendation to improve the article:

For lines 65-67, Please mention a couple of lines about the implications of this. Also, for line 84, mention briefly, based on the literature, the known mechanism by which tamoxifen is a risk for other cancers.

I liked the tables that the authors presented to summarize the CDK 4/6 inhibitor PI3K/Akt/mTOR pathway inhibitors. It would be great if the authors could also summarize SERMs, aromatase inhibitors, and SERD in tables. These types of tables are always helpful for the researchers and clinicians to have a quick look.

Author Response

Reviewer 1

In this review article, the authors thoroughly described the available targeted therapy approaches in the clinic to deal with the Hormone Receptor-Positive and HER2-negative breast cancer (HR+BC), the most notorious subtype of breast cancer. Here is the minor recommendation to improve the article: For lines 65-67, please mention a couple of lines about the implications of this.

We have added additional text highlighting the implications of differential effects of tamoxifen in various tissues in lines 71-77 (track changes).

Also, for line 84, mention briefly, based on the literature, the known mechanism by which tamoxifen is a risk for other cancers.

We have specified it earlier in the SERMS section in lines 77-79

I liked the tables that the authors presented to summarize the CDK 4/6 inhibitor PI3K/Akt/mTOR pathway inhibitors. It would be great if the authors could also summarize SERMs, aromatase inhibitors, and SERD in tables. These types of tables are always helpful for the researchers and clinicians to have a quick look.

We appreciate the suggestion by the reviewer and have added a new table (Table 2) highlighting the key data of antiestrogen treatments in women with early stage breast cancer or those who are at high risk of development of breast cancer.

Reviewer 2 Report

Authors provide a comprehensive background of treatment in hormone positive breast cancers, Few things they can consider addressing 

What are the limitations of certain treatment options 

Sometimes Prophylactic ovary removal is considered in premenopausal women which have aggressive cancer, can authors throw some light on that?

Can authors list more current clinical trials on investigational agents which are currently evaluated in late phase clinical trials?

Author Response

Reviewer 2

Authors provide a comprehensive background of treatment in hormone positive breast cancers, few things they can consider addressing 

What are the limitations of certain treatment options? 

Limitation of various treatment options are highlighted throughout the text as outlined below

  • lines 94-96 limitations of chemoprevention
  • lines 110-112 lack of overall survival advantages of antiestrogen treatment in DCIS
  • lines 138-140 limitations of extended tamoxifen
  • lines 214-218 limitations of combination endocrine therapy in premenopausal women
  • lines 284-288 limitations of combination of fulvestrant and an AI
  • lines 416-418 limitations of adjuvant CDK4/6 inhibitors
  • lines 475-475 limitations of everolimus in previously treated patients with AI and CDK4/6 inhibitors
  • lines 487-88, 501-502 limitations of PI3KCA inhibitors
  • lines 525-26 limitations of histone deacetylase inhibitor
  • lines 539-40 limitations of anti-VEGF
  • lines 636-37 about resistance

Sometimes Prophylactic ovary removal is considered in premenopausal women which have aggressive cancer, can authors throw some light on that?

We have provided data from SOFT and TEXT trial that ovarian suppression in combination with tamoxifen or AI in younger women or high risk disease has been associated with better outcomes in lines 209-13 and have added ovary removal as an option in lines 207-209

Can authors list more current clinical trials on investigational agents which are currently evaluated in late phase clinical trials?

A paragraph highlighting several novel targeted compounds/strategies have been added in the revised manuscript (lines 600-609 and 614-616)

Reviewer 3 Report

  1. 3 line 98 There are no data about survival benefit or there is no survival benefit shown in trials?
  2. 3 line 112: I suggest to add the reference of the TAILORx trial
  3. 3 line 125: why a valid option only in premopausal women? There are some women that do not tolerate AI and could benefit from extended tamoxifen
  4. 3 line 129: I suggest to add: “in post-menopausal women”
  5. 4 line 177: Title “adjuvant AI vs AI” not descriptive enough
  6. 5 line 197: Suggest detailing which toxicities are higher.

P 5: section 3.2.2.2.5: suggest adding data about OS

  1. 6 section 3.3.2: Was the benefit seen only in de novo metastatic breast cancer?
  2. 6: table 1 would be useful earlier in the text
  3. 7 line 320: who progressed on “adjuvant” or “1st line” endocrine therapy? Please clarify for the readers.

Table 3 : « line of estrogen therapy » Should it be called as “line of treatment”?

Also, it could be 1st line in certain circumstance (ex. As stated p. 8 line 331, for some women that relapsed within 12 months of completing adjuvant endocrine therapy).

  1. 9, paragraph 383. It would be interesting to discuss that the curves were initially diverging and after they came back together. This is the question with MonarchE with longer follow-up if it will do the same thing.

Overall, suggesting putting p values in the tables.

  1. 12 line 490 Authors could mention that the FDA indication for MBC bevacizumab has been withdrawn.
  2. 12, “Other compounds”. Why not mention the EMBRACA trial-talazoparib?
  3. 13. Figure 1. Typo in “Pot-menopausal” and “ologometasttic” Don’t think that figure should be there. After progression on CDK not sure what is the best line. Also, something we give chemo if endocrine resistant so without a discussion about guidelines, suggest removing the figure.

Author Response

Reviewer 3

Line 98 There are no data about survival benefit or there is no survival benefit shown in trials?

It has been specified in lines 110-112 that adjuvant tamoxifen in DCIS does not prolong overall survival. 

Line 112: I suggest to add the reference of the TAILORx trial

A reference to TAILORx trial has been added (reference 28).

Line 125: why a valid option only in premopausal women? There are some women that do not tolerate AI and could benefit from extended tamoxifen

We agree with the reviewer comment and have omitted ‘premenopausal’ as tamoxifen is an option for both premenopausal and post-menopausal women who could not tolerate an AI.

Line 129: I suggest to add: “in post-menopausal women”

We appreciate this suggestion. However, the title reflects a class of medicine. Therefore, we have not added it in the title but have specified in the text in lines 141-42.

Line 177: Title “adjuvant AI vs AI” not descriptive enough

We have modified the title to “steroidal AI vs no-steroidal AI”

Line 197: Suggest detailing which toxicities are higher.

It has been added in the revised manuscript in the lines 214-218

Section 3.2.2.2.5: suggest adding data about OS

As per reviewer suggestion it has been elaborated further in the lines 226-232

Section 3.3.2: Was the benefit seen only in de novo metastatic breast cancer?

It has been added that the benefit was significant in women who did not receive adjuvant tamoxifen in lines 284-88

Table 1 would be useful earlier in the text

Table 1 has been moved early in the text

line 320: who progressed on “adjuvant” or “1st line” endocrine therapy? Please clarify for the readers.

It has been clarified.

Table 3 : « line of estrogen therapy » Should it be called as “line of treatment”? Also, it could be 1st line in certain circumstance (ex. As stated p. 8 line 331, for some women that relapsed within 12 months of completing adjuvant endocrine therapy).

It has been changed to line endocrine therapy.

9, paragraph 383. It would be interesting to discuss that the curves were initially diverging and after they came back together. This is the question with MonarchE with longer follow-up if it will do the same thing.

Reference to survival curves has been added in the lines 416-418

Overall, suggesting putting p values in the tables.

In most cases 95% confidence intervals have been provided. In addition, p values have been added when available.

Line 490 Authors could mention that the FDA indication for MBC bevacizumab has been withdrawn.

It has been added in line 534-36.

 “Other compounds”. Why not mention the EMBRACA trial-talazoparib?

We appreciate reviewer comments and have add information about talazoparib in lines 551-557

Figure 1. Typo in “Pot-menopausal” and “ologometasttic” Don’t think that figure should be there. After progression on CDK not sure what is the best line. Also, something we give chemo if endocrine resistant so without a discussion about guidelines, suggest removing the figure.

As per reviewer suggestion we have deleted figure 1 from the revised manuscript.
